# Carbonic Anhydrase Inhibition as a Target for Antibiotic Synergy in Enterococci

Gayatri Shankar Chilambi,[a]* Yu-Hao Wang,[a] Nathan R. Wallace,[a] Chetachukwu Obiwuma,[a] Kirsten M. Evans,[a] Yanhong Li,[a,b] Menna-Allah W. Shalaby,[c] Daniel P. Flaherty,[c] Ryan K. Shields,[a] Yohei Doi,[a] Daria Van Tyne[a]

[a]Division of Infectious Diseases, University of Pittsburgh School of Medicine, Pittsburgh, Pennsylvania, USA
[b]Tsinghua University School of Medicine, Beijing, China
[c]Department of Medicinal Chemistry and Molecular Pharmacology, Purdue University, West Lafayette, Indiana, USA

**ABSTRACT** *Enterococcus faecalis* is a hospital-associated opportunistic pathogen that can cause infections with high mortality, such as infective endocarditis. With an increasing occurrence of multidrug-resistant enterococci, there is a need for alternative strategies to treat enterococcal infections. We isolated a gentamicin-hypersusceptible *E. faecalis* strain from a patient with infective endocarditis that carried a mutation in the alpha-carbonic anhydrase ($\alpha$-CA) and investigated how disruption of $\alpha$-CA sensitized *E. faecalis* to killing with gentamicin. The gentamicin-hypersusceptible $\alpha$-CA mutant strain showed increased intracellular gentamicin uptake in comparison to an isogenic strain encoding full-length, wild-type $\alpha$-CA. We hypothesized that increased gentamicin uptake could be due to increased proton motive force (PMF), increased membrane permeability, or both. We observed increased intracellular ATP production in the $\alpha$-CA mutant strain, suggesting increased PMF-driven gentamicin uptake contributed to the strain's gentamicin susceptibility. We also analyzed the membrane permeability and fatty acid composition of isogenic wild-type and $\alpha$-CA mutant strains and found that the mutant displayed a membrane composition that was consistent with increased membrane permeability. Finally, we observed that exposure to the FDA-approved $\alpha$-CA inhibitor acetazolamide lowered the gentamicin MIC of eight genetically diverse *E. faecalis* strains with intact $\alpha$-CA but did not change the MIC of the $\alpha$-CA mutant strain. These results suggest that $\alpha$-CA mutation or inhibition increases PMF and alters membrane permeability, leading to increased uptake of gentamicin into *E. faecalis*. This connection could be exploited clinically to provide new combination therapies for patients with enterococcal infections.

**IMPORTANCE** Enterococcal infections can be difficult to treat, and new therapeutic approaches are needed. In studying an *E. faecalis* clinical strain from an infected patient, we found that the bacteria were rendered hypersusceptible to aminoglycoside antibiotics through a mutation that disrupted the $\alpha$-CA. Our follow-on work suggested two different ways that $\alpha$-CA disruption causes increased gentamicin accumulation in *E. faecalis*: increased proton motive force-powered uptake and increased membrane permeability. We also found that a mammalian CA inhibitor could sensitize a variety of *E. faecalis* strains to killing with gentamicin. Given that mammalian CA inhibitors are frequently used to treat conditions such as glaucoma, hypertension, and epilepsy, our findings suggest that these "off-the-shelf" inhibitors could also be useful partner antibiotics for the treatment of *E. faecalis* infections.

**KEYWORDS** carbonic anhydrase, antibiotic synergy, *Enterococcus faecalis*

*E*nterococcus faecalis is a Gram-positive facultative anaerobe that resides in the human gastrointestinal tract. *E. faecalis* is also a sporadic pathogen that causes opportunistic and hospital-associated infections (1). In particular, *E. faecalis* causes infective endocarditis (EFIE) which is associated with mortality rates as high as 30%. EFIE mortality rates have

Address correspondence to Daria Van Tyne, vantyne@pitt.edu.

*Present address: Gayatri Shankar Chilambi, Public Health Research Institute and Department of Medicine, Rutgers New Jersey Medical School, Newark, New Jersey, USA.

The authors declare no conflict of interest.

**TABLE 1** MICs of different *E. faecalis* $\alpha$-CA genotypes[a]

| Strain ID | Source | MIC ($\mu$g/mL) | | | | | | $\alpha$-CA allele | |
| | | GEN | STREP | TOB | AMP | TET | CIP | Codon at position 209 | Amino acid |
|---|---|---|---|---|---|---|---|---|---|
| DVT809 | Infective endocarditis | 1 | 8 | 1 | 1 | 0.5 | 2 | TAA | 209* |
| DVT1133 | *In vitro* evolved derivative of DVT809 | 16 | 128 | 16 | 1 | 0.5 | 4 | TTA | 209Leu |
| DVT1134 | *In vitro* evolved derivative of DVT809 | 16 | 128 | 16 | 1 | 0.5 | 4 | GAA | 209Glu (WT) |
| DVT1135 | *In vitro* evolved derivative of DVT809 | 16 | 128 | 16 | 1 | 0.5 | 4 | TAT | 209Tyr |

[a]ID, identifier; GEN, gentamicin; STREP, streptomycin; TOB, tobramycin; AMP, ampicillin; TET, tetracycline; CIP, ciprofloxacin; WT, wild type.

remained largely unchanged for the last 30 years, despite new antimicrobial therapies (2, 3). Combination antibiotic therapy is the preferred treatment option for EFIE (4–6); however, the emergence of strains that are resistant to these agents has prompted a need to explore alternative treatment options.

Carbonic anhydrase (CA) enzymes are responsible for the hydration of carbon dioxide to produce protons and bicarbonate ($HCO_3^-$) in living organisms. Nucleic acid, fatty acid, and amino acid biosynthetic pathways contain an essential $HCO_3^-$-dependent carboxylation step, and CAs provide $CO_2/HCO_3^-$ for these enzymatic reactions (7, 8). Bacterial CAs have recently been identified as novel antibacterial targets (9, 10). For this reason, CA inhibitors have been previously shown to reduce the virulence of bacteria such as *Vibrio cholerae*, *Mycobacterium tuberculosis*, and *Helicobacter pylori* (11–13). Moreover, FDA-approved CA inhibitors have been identified as promising antienterococcal targets (14, 15). While prior studies have found that bacteria can possess $\alpha$-, $\beta$-, and $\gamma$-CAs (9), *E. faecalis* genomes appear to encode only enzymes belonging to the $\alpha$- and $\gamma$-CA families.

Here, we describe a gentamicin-hypersusceptible strain of *E. faecalis* that was isolated from a patient with EFIE at our medical center. To determine the basis of gentamicin hypersusceptibility, we performed whole-genome sequencing of the clinical strain as well as *in vitro* evolved one-step mutants with restored intrinsic gentamicin resistance. This analysis revealed that gentamicin hypersusceptibility in the original clinical strain was caused by a mutation that disrupted the *E. faecalis* $\alpha$-CA and led us to hypothesize that disruption of the $\alpha$-CA could sensitize *E. faecalis* to killing with aminoglycosides. We investigated possible mechanisms responsible and found evidence supporting both active uptake and passive diffusion as causing increased intracellular antibiotic concentration in the context of $\alpha$-CA disruption.

## RESULTS

**$\alpha$-CA mutation causes gentamicin hypersusceptibility in an *E. faecalis* clinical strain.** While characterizing *E. faecalis* strains from patients with infective endocarditis (EFIE), we identified a clinical strain, called DVT809, which was found to be hypersusceptible to gentamicin. DVT809 had a gentamicin MIC of 1 $\mu$g/mL (Table 1), which was well below the intrinsic low-level gentamicin resistance normally exhibited by enterococci (16). To determine whether gentamicin hypersusceptibility in this strain was due to a single point mutation, we conducted a 1-step *in vitro* evolution experiment by plating DVT809 onto agar plates containing 4 $\mu$g/mL of gentamicin. We isolated three mutant strains (DVT1133, DVT1134, and DVT1135) from independent experiments that each demonstrated a 16-fold increase in gentamicin MIC compared to the parent DVT809 strain (Table 1). Suspecting that these mutants might have reverted a gentamicin hypersusceptibility-causing mutation, we sequenced their genomes and compared them to the genome of DVT809. We found that all three mutants carried single-base changes at codon 209 of the alpha-carbonic anhydrase ($\alpha$-CA) (Table 1). Comparison of the $\alpha$-CA sequence in the hypersusceptible parent and reverted derivatives, as well as with other *E. faecalis* genome sequences in the NCBI database, revealed that the hypersusceptible DVT809 strain carried a unique mutation causing a premature stop codon at position 209 (209*) (Table 1). A homology model of the *E. faecalis* $\alpha$-CA was then constructed using the structure from *Thermovibrio ammonificans* (PDB 4COQ) (17) as a template. The model showed that protein truncation after residue 209 would remove a C-terminal strand running nearly halfway around the enzyme with side chains and the backbone making numerous stabilizing interactions to each adjacent strand. It is reasonable to hypothesize that truncation of this strand may result in destabilization of

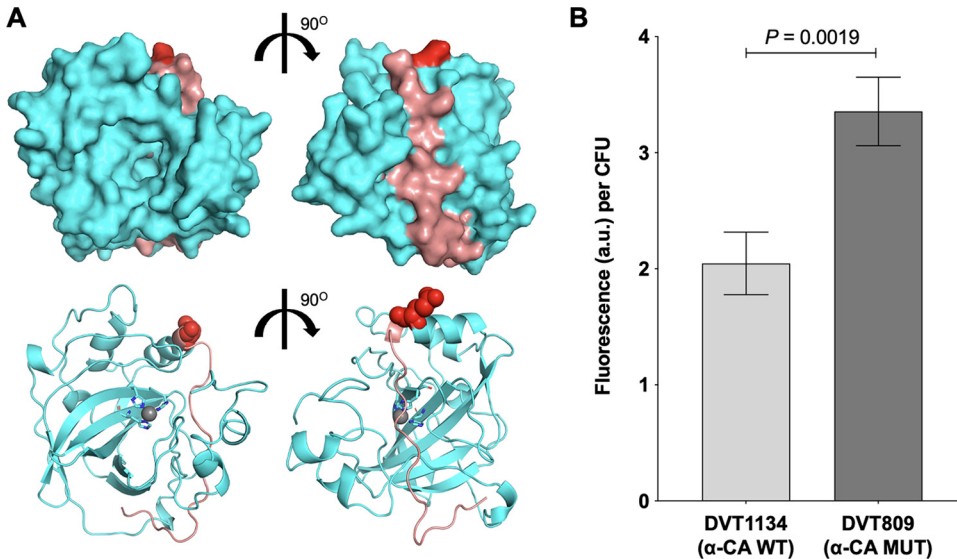

**FIG 1** Alpha-carbonic anhydrase ($\alpha$-CA) disruption is associated with increased gentamicin-Texas Red (GTTR) uptake in *E. faecalis*. (A) Surface (top) and ribbon (bottom) homology model views of the *E. faecalis* $\alpha$-CA. Residue E209 is colored red, and residues that sit C-terminal to E209—which are predicted to be missing in the DVT809 $\alpha$-CA mutant strain—are colored salmon. (B) GTTR uptake into DVT1134 $\alpha$-CA wild-type (WT) or DVT809 $\alpha$-CA mutant (MUT) strain. Bars show mean fluorescence intensity values (arbitrary units, a.u.) per CFU, and error bars denote standard error of the mean. CFU present at the beginning of the experiment was used to normalize fluorescence intensities for each strain. *P* value is from a two-tailed *t* test.

the quaternary complex and yield an inactive enzyme; however, further tests would be necessary to confirm this hypothesis.

All three one-step mutants encoded full-length $\alpha$-CA sequences but had different residues at position 209 (Table 1). The $\alpha$-CA sequence of the DVT1134 strain, which encoded a glutamate at position 209 (209Glu), was the most prevalent sequence among *E. faecalis* genomes in the NCBI database; thus, we assigned this strain the wild-type genotype. No additional mutations were found in DVT1134 compared to the DVT809 parent strain; thus, these two strains represented an isogenic wild-type and mutant strain pair. We also investigated the susceptibility of the DVT809 parent and reverted mutant strains to other aminoglycosides, as well as antibiotics belonging to different classes (Table 1). We observed that DVT809 was also more susceptible to streptomycin and tobramycin than were DVT1133, DVT1134, and DVT1135. However, all strains had similar MICs of ampicillin (MIC = 1 $\mu$g/mL), ciprofloxacin (MIC = 2 to 4 $\mu$g/mL), and tetracycline (MIC = 0.5 $\mu$g/mL) (Table 1). These data suggest that $\alpha$-CA truncation in the DVT809 strain causes hypersusceptibility to aminoglycosides but not to antibiotics in other classes.

**Increased gentamicin accumulation is associated with $\alpha$-CA disruption.** Enterococci are intrinsically resistant to clinically achievable concentrations of aminoglycosides, which has been attributed to decreased uptake of these compounds (18). Previous studies have used fluorescently labeled aminoglycosides to assess differences in antibiotic uptake (19, 20). We compared gentamicin uptake in the hypersusceptible DVT809 and wild-type DVT1134 strains using gentamicin-Texas Red (GTTR) (Fig. 1). Because we observed differences in the *in vitro* growth rates of the DVT809 and DVT1134 strains suggesting that DVT809 cells might have compromised fitness (see Fig. S1 in the supplemental material), we measured intracellular fluorescence following incubation with GTTR and normalized the fluorescent signal to the number of CFU of bacteria. We found that the hypersusceptible DVT809 strain showed significantly more intracellular GTTR fluorescence than the DVT1134 strain encoding a wild-type $\alpha$-CA (Fig. 1) (*P* = 0.0019). This result suggests that the gentamicin hypersusceptibility of the DVT809 strain is due to increased gentamicin uptake.

**Increased intracellular ATP accumulation is associated with $\alpha$-CA disruption and gentamicin susceptibility in *E. faecalis*.** Aminoglycoside uptake is an energy-dependent process that requires both an electrochemical gradient and electron flow through the electron transport chain (21). The normally anaerobic metabolism of enterococci is thought

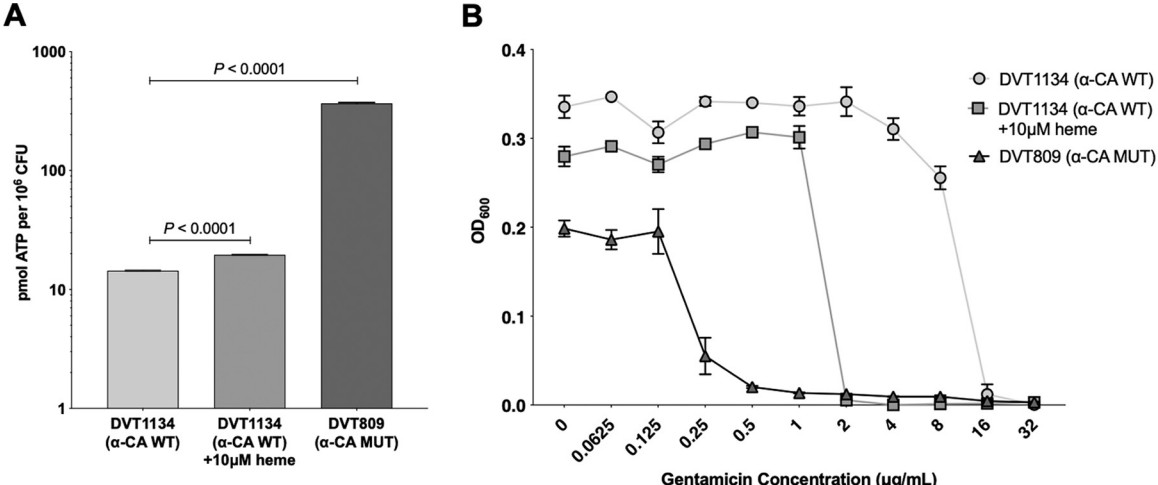

**FIG 2** Increased intracellular ATP accumulation and increased PMF are associated with gentamicin susceptibility in *E. faecalis*. (A) Intracellular ATP measured in mid-log-phase cultures of DVT1134 (wild type, WT), DVT1134 grown in the presence of 10 $\mu$M heme, and DVT809 ($\alpha$-CA mutant, MUT). Bars show mean picomoles of ATP per million CFU, and error bars denote standard error of the mean. CFU present at the beginning of the experiment was used to normalize ATP abundance. *P* values are from two-tailed *t* tests. (B) Growth of strains shown in panel A in various concentrations of gentamicin. Optical densities measured at 600 nm (OD$_{600}$) versus gentamicin concentration are shown. Data points show the averages from three biological replicates, and error bars represent the standard error of the mean.

to contribute to their intrinsic aminoglycoside resistance by limiting drug uptake (22). We wondered whether the increased gentamicin uptake observed in the DVT809 strain was due to increased proton motive force (PMF)-powered transport. We quantified intracellular ATP concentrations in the DVT1134 wild-type and DVT809 $\alpha$-CA mutant strains and observed that DVT809 contained significantly more intracellular ATP (Fig. 2A) ($P <$ 0.0001). In the presence of heme, enterococci can form a functional electron transport chain and increase their PMF (23). We confirmed that incubation with heme increased intracellular ATP production in the DVT1134 wild-type strain ($P <$ 0.0001) (Fig. 2A). Exposure to heme also increased gentamicin susceptibility in the wild-type DVT1134 strain but not to the same level of hypersusceptibility as that of the DVT809 strain (Fig. 2B). Taken together, these results suggest that increased ATP production and greater PMF contribute to gentamicin hypersusceptibility in DVT809.

**Altered membrane permeability and fatty acid composition are associated with $\alpha$-CA disruption in *E. faecalis*.** Several studies have shown that aminoglycoside activity can be increased by destabilizing the bacterial membrane with antimicrobial lipids, antimicrobial peptides, and retinoids (24–27). To test whether $\alpha$-CA disruption caused altered membrane permeability in *E. faecalis*, we assessed the membrane integrity of the DVT1134 wild-type and DVT809 $\alpha$-CA mutant strains using propidium iodide (PI), a DNA-binding fluorophore that penetrates cells with compromised membranes (28, 29). We observed that DVT809 showed increased PI fluorescence compared to DVT1134 (Fig. 3A), suggesting that DVT809 cell membranes were more permeable to small molecules. To understand the nature of this increased membrane permeability, we quantified the membrane composition of the DVT1134 and DVT809 strains with fatty acid methyl ester (FAME) analysis. The membranes of both strains were composed primarily of palmitic acid (a saturated fatty acid), palmitoleic and *cis*-vaccenic acids (unsaturated fatty acids), and cyclopropane fatty acid (Fig. 3B). Compared to DVT1134, the DVT809 membrane was composed of significantly less palmitic and cyclopropane fatty acids, and significantly more *cis*-vaccenic acid, an unsaturated fatty acid. While differences in membrane fatty acid content can impact a variety of cellular functions, the differences we observed are consistent with DVT809 having a more fluid and permeable membrane. Together these results suggest that altered membrane composition and increased PMF likely contribute to the increased gentamicin uptake observed in this strain.

**The mammalian CA inhibitor acetazolamide sensitizes *E. faecalis* with intact $\alpha$-CA to killing with gentamicin.** Acetazolamide is a broad-range CA inhibitor that has been shown to be effective against several bacterial species, including clinical isolates of vancomycin-resistant *Enterococcus* (VRE) (30). We tested whether acetazolamide coincubation could

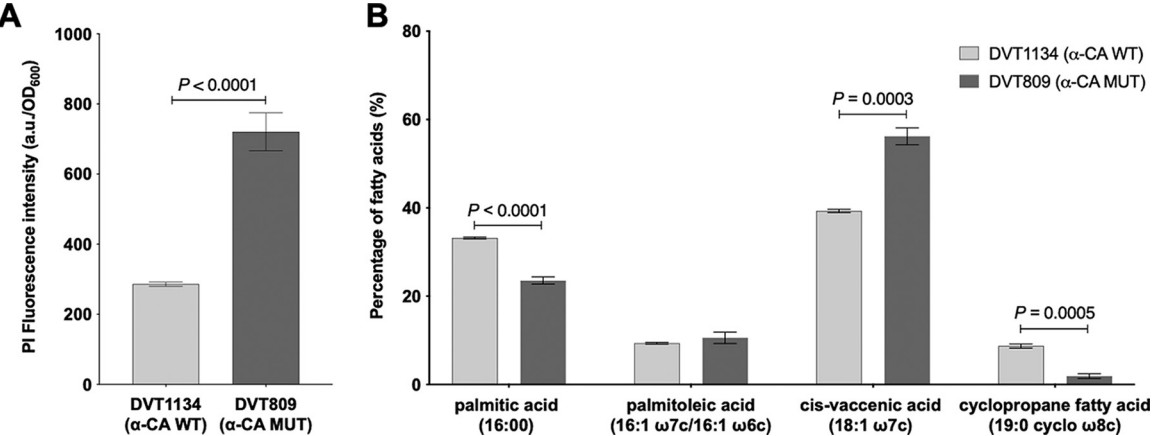

**FIG 3** Differences in membrane permeability and fatty acid composition associated with $\alpha$-CA disruption in *E. faecalis*. (A) Propidium iodide (PI) uptake into mid-log-phase cultures of DVT1134 (wild type, WT) and DVT809 ($\alpha$-CA mutant, MUT) strains. Bars show mean fluorescence intensity values (arbitrary units, a.u.) of six biological replicates and three technical replicates, normalized to the optical density of the culture measured at 600 nm ($OD_{600}$). Error bars denote standard error of the mean. (B) Fatty acid methyl ester (FAME) abundance of fatty acids comprising more than 10% of the total membrane fatty acid content of the DVT1134 and DVT809 strains. Bars show mean percentages from four biological replicates, and error bars denote standard error of the mean. All *P* values are from two-tailed *t* tests.

sensitize *E. faecalis* to the bactericidal effects of gentamicin by testing a panel of genetically diverse *E. faecalis* strains including DVT809 and eight other strains predicted to have an intact $\alpha$-CA, including six strains collected from patients with *E. faecalis* endophthalmitis (31). We observed synergy between acetazolamide and gentamicin in all cases except DVT809, where coincubation with 2 $\mu$g/mL acetazolamide caused an 8- to 64-fold drop in the gentamicin MIC (Table 2). These data indicate that $\alpha$-CA inhibition sensitizes *E. faecalis* to killing with gentamicin.

## DISCUSSION

The objective of this study was to explore how disruption of the $\alpha$-CA sensitizes *E. faecalis* to killing with gentamicin. We used comparative genomics, measurements of antibiotic uptake, and membrane transport and permeability assays to test whether in the absence of a functional $\alpha$-CA, *E. faecalis* takes up more gentamicin due to increased PMF-driven transport as well as enhanced membrane permeability. Our findings support the potential of CA inhibitors as new therapeutic partners for the treatment of antibiotic-resistant enterococcal infections.

Prior studies have shown that enterococci are intrinsically resistant to aminoglycosides due to poor antibiotic uptake caused by their facultative anaerobic metabolism (22, 32). Consistent with prior literature, we found that gentamicin hypersusceptibility was correlated with increased gentamicin uptake in a single *E. faecalis* strain with an inactivating mutation in the $\alpha$-CA. The strain showed hypersusceptibility to other aminoglycosides besides gentamicin but not to tetracycline or ampicillin. The strain also showed evidence of

**TABLE 2** MIC of gentamicin in the absence and presence of 2 $\mu$g/mL acetazolamide against genetically diverse *E. faecalis* strains[a]

| Strain ID | Sequence type (ST) | MIC ($\mu$g/mL) | | Fold sensitization |
|---|---|---|---|---|
| | | GEN | GEN + 2 $\mu$g/mL AZA | |
| OG1RF | 1 | 32 | 4 | 8 |
| DVT809 | 482 | 1 | 1 | 1 |
| DVT1134 | 482 | 16 | 0.5 | 32 |
| E46 | 64 | 16 | 0.25 | 64 |
| E263 | 55 | 16 | 0.25 | 64 |
| E266 | 25 | 16 | 1 | 16 |
| E616 | 122 | 16 | 0.25 | 64 |
| E676 | 428 | 8 | 0.5 | 16 |
| E687 | 34 | 8 | 1 | 8 |

[a]ST, multilocus sequence type; GEN, gentamicin; AZA, acetazolamide; ID, identifier.

increased PMF-powered transport compared to an isogenic wild-type strain, which could be one cause of increased antibiotic uptake. Taken together, these findings suggest that a mutated $\alpha$-CA causes a deficit of protons on the cytoplasmic side of the bacterial membrane, leading to increased $F_oF_1$-ATPase activity, which in turn drives the uptake of gentamicin and other molecules into the cell.

In addition to active transport, small molecules can also enter bacteria by passing directly through the cell membrane. It was recently shown that rhamnolipids can stimulate PMF-independent aminoglycoside uptake in *Staphylococcus aureus* by altering membrane permeability, membrane fluidity, and cell surface charge (20). Similarly, synergy has been previously observed between aminoglycosides and membrane-interacting antimicrobial lipids, antimicrobial peptides, and retinoids (24–27); however, the mechanism underlying this synergy is unclear. In our study, we found a higher relative abundance of unsaturated fatty acids, specifically *cis*-vaccenic acid, in the gentamicin-hypersusceptible $\alpha$-CA mutant DVT809 strain. The presence of more unsaturated fatty acids could make the membrane more fluid or permeable in structure (33), which might allow for increased passage of molecules like gentamicin into the cell. Alternately or in addition, differences in membrane composition might alter the function of membrane proteins and thereby exert an indirect effect on aminoglycoside uptake. The mechanistic connection between $\alpha$-CA disruption and altered membrane composition is currently unknown and will be the focus of our future work in this area.

CA inhibitors constitute a promising new class of potential antimicrobials for several reasons. First, they have been shown to inhibit the growth of a wide variety of bacteria (9, 10) and to synergize with other antibiotics to inhibit enterococci specifically (14). Second, several FDA-approved CA inhibitors with known activity against enterococci already exist. One of these inhibitors, acetazolamide, has been used as a scaffold to develop optimized inhibitors of vancomycin-resistant enterococci (30). Consistent with these prior reports, we found that acetazolamide could synergize with gentamicin against genetically distinct *E. faecalis* clinical strains with intact $\alpha$-CA proteins; however, the degree of synergy was somewhat variable between strains. In the future we plan to explore the contribution of $\alpha$-CA sequence diversity to this synergy, as well as other potential mediators of CA activity.

This study had several limitations. We identified a single *E. faecalis* clinical strain with a disrupted $\alpha$-CA, and it is unknown whether $\alpha$-CA disruption is a more general phenomenon among clinical *E. faecalis* strains. Our characterization of the gentamicin hypersusceptibility in this strain was limited to *in vitro* assays and aerobic bacterial growth, and we did not make direct microscopic measurements of gentamicin uptake in the $\alpha$-CA mutant or wild-type strains. Furthermore, we examined gentamicin uptake in stationary-phase cultures (which are less susceptible to gentamicin killing) but measured ATP production and membrane permeability in exponential-phase cultures. Additionally, our observations were limited to disruption of the $\alpha$-CA, and we did not investigate the *E. faecalis* $\gamma$-CA. Finally, the precise molecular links between $\alpha$-CA disruption, increased PMF-driven antibiotic uptake, and increased membrane permeability remain unknown.

Overall, we find that $\alpha$-CA disruption likely sensitizes *E. faecalis* to the bactericidal effects of gentamicin due to increased PMF-powered uptake as well as increased membrane permeability. We also found that acetazolamide, a mammalian CA inhibitor, rendered genetically diverse *E. faecalis* strains more susceptible to gentamicin inhibition. Because several CA inhibitors with antibiotic activity are already FDA approved, our study suggests that these inhibitors could be leveraged as new partner molecules for the treatment of *E. faecalis* infections.

## MATERIALS AND METHODS

**Bacterial growth, whole-genome sequencing, and analysis.** Bacteria were propagated in brain heart infusion (BHI) medium or Mueller-Hinton broth (MHB) as indicated for each experiment below and were grown at 37°C with shaking at 170 rpm. Genomic DNA from *E. faecalis* strains was extracted using a DNeasy Blood and Tissue kit (Qiagen, Germantown, MD) from 1-mL bacterial cultures grown in BHI medium. Next-generation sequencing libraries were prepared with a Nextera kit (Illumina, San Diego, CA) and were sequenced on an Illumina NextSeq550 sequencer using 150-bp paired-end reads. Genomes were assembled with SPAdes v3.13.0 (34) and were annotated with Prokka (35). Variants were identified using CLC Genomics Workbench

v11.0.1 (Qiagen, Germantown, MD), using a read depth cutoff of 10 reads and a variant frequency cutoff of >90%.

**Homology modeling of *E. faecalis* $\alpha$-CA.** The SWISS-Model web server (36) was used to construct a homology model for the *E. faecalis* $\alpha$-CA. Upon submission of the target protein sequence in FASTA format, the automated pipeline employed BLAST and HHblits to identify the appropriate templates from the SWISS Model Template Library (SMTL; version 2023-03-08) (37, 38). From the list of the templates provided, which included pertinent structural data, the $\alpha$-CA structure from *Thermovibrio ammonificans* (PDB 4COQ) (17) was selected as the template due to sequence identity (34.4%), quality of the structure (solved to 1.5-Å resolution), and quality of the preliminary homology model. The homology model was constructed using ProMod3 (39), which incorporates four steps in the modeling process: template superposition, alignment of the target and the template, model construction, and energy minimization. The model was evaluated using the QMEAN scoring function to assess both the overall quality of the model and the quality of each residue individually (40, 41). The homology model for *E. faecalis* $\alpha$-CA that was generated exhibited a global model quality estimate (GMQE) of 0.71 and a QMEANDisCo of 0.07 $\pm$ 0.06. To further validate the model, it was aligned with the template structure (PDB code 4COQ) using PyMOL, and we obtained a root mean square deviation (RMSD) value of 0.18 Å (RMSD values of ≤2 Å are considered good; smaller values imply greater similarity).

**Antimicrobial susceptibility testing.** Antibiotic susceptibility testing to determine the MICs of ampicillin, ciprofloxacin, gentamicin, streptomycin, tetracycline, and tobramycin was carried out by broth microdilution in cation-adjusted Mueller-Hinton broth (MHB) according to methods established by the Clinical and Laboratory Standards Institute (42, 43). Briefly, overnight cultures of *E. faecalis* grown in MHB were diluted to an optical density at 600 nm ($OD_{600}$) of 0.1 and were further diluted 1:1,000 into fresh MHB. One hundred microliters of this culture was then transferred to 96-well plates containing 100 $\mu$L of MHB with serial 2-fold dilutions of each antibiotic, yielding approximately $10^5$ bacteria per well. Plates were incubated for 24 h at 37°C under static conditions, and growth in each well was analyzed both by visual inspection and by $OD_{600}$ measurement using a Synergy H1 microplate reader (BioTek, Winooski, VT). Gentamicin susceptibility was also determined in the presence of 10 $\mu$M heme and 2 $\mu$g/mL of acetazolamide. Experiments were performed in triplicate with three biological replicates and three technical replicates, and the mode MIC value for each experiment was reported.

**Gentamicin-Texas Red uptake.** Gentamicin-Texas Red (GTTR) was prepared as previously described (44). Briefly, Texas Red-succinimidyl ester (Invitrogen, Waltham, MA) was dissolved in high-quality anhydrous *N,N*-dimethylformamide at 4°C at a final concentration of 20 mg/mL. Gentamicin was dissolved in 100 mM $K_2CO_3$, pH 8.5, at a final concentration of 10 mg/mL. At 4°C, 10 $\mu$L of Texas Red was slowly added to 350 $\mu$L gentamicin solution to allow a conjugation reaction to proceed. The reaction mixture components were agitated together at 4°C for 3 days to produce GTTR. Unbound Texas Red from the reaction mixture was removed with Pierce dye removal columns (Thermo Scientific, Waltham, MA) following the manufacturer's instructions (19). Overnight cultures of the $\alpha$-CA mutant DVT809 and wild-type DVT1134 strains were normalized to an $OD_{600}$ of 0.5, and 1 mL of each culture was washed twice with phosphate-buffered saline (PBS) and then incubated in PBS with GTTR at a final concentration of 2.5 $\mu$g/mL at 37°C. After 30 min, the supernatant was discarded and the cells were washed in PBS prior to being transferred to Power Bead tubes (Qiagen) for cell lysis according to the manufacturer's protocol. The supernatant of the lysed cells was centrifuged at 13,000 rpm for 2 min and then analyzed by measuring fluorescence at excitation/emission wavelengths of 510/610 nm, respectively, on a Synergy H1 microplate reader (BioTek, Winooski, VT). Aliquots of each culture were taken prior to GTTR exposure and were used to enumerate CFU per milliliter for data normalization. Experiments were performed in triplicate with three biological replicates.

**Determination of intracellular ATP levels.** Exponential-phase cultures of $\alpha$-CA mutant DVT809 and wild-type DVT1134 strains were centrifuged at 13,000 rpm for 2 min, and supernatants were collected for ATP measurement. Briefly, 100 $\mu$L of culture supernatants or uncentrifuged cell cultures was mixed with equal volumes of BacTiter-Glo ATP measurement reagent (Promega, Inc., WI), and luminescence was measured after 5 min with a Synergy H1 microplate reader (BioTek, Winooski, VT). Luminescence values were compared between supernatants (extracellular ATP) and uncentrifuged cell cultures (total microbial ATP). A standard curve was used to determine ATP quantities in each sample. ATP abundance was normalized to CFU present prior to treatment, and values are reported as picomoles of ATP per million CFU. Experiments were performed in triplicate with three biological replicates.

**PI uptake.** Exponential-phase cultures of *E. faecalis* strains DVT809 and DVT1134 were washed and resuspended in PBS to an $OD_{600}$ of 0.5 and were then incubated with 1$\times$ PBS for 5 min. After incubation, cells were pelleted at 6,500 rpm for 5 min and were washed twice in PBS, followed by incubation with 20 $\mu$g/mL propidium iodide (PI) for 5 min. PI-treated cells were washed and resuspended in PBS, and then fluorescence intensities were quantified at excitation/emission wavelengths of 535/617 nm, respectively, on a Synergy H1 microplate reader (BioTek, Winooski, VT). Experiments were performed in triplicate with three biological replicates.

**FAME analysis.** Bacterial cultures were grown to mid-log phase ($OD_{600}$ of 0.5) in 25 mL of BHI broth with shaking at 37°C. Cells were harvested via centrifugation (4,000 rpm, 10 min), washed twice with 1$\times$ PBS, pelleted, and stored at $-20$°C before being shipped to Microbial ID (Newark, DE) for fatty acid methyl ester (FAME) analysis (45). Briefly, cells were subjected to saponification with sodium hydroxide and methanol prior to the addition of hydrochloric acid-methyl alcohol to cause fatty acid methylation. Fatty acid methyl esters were then extracted with hexane and methyl *tert*-butyl ether prior to FAME analysis by gas chromatography.

**Statistical analysis.** Two-tailed Student's *t* tests were used to compare data between different strains and conditions in each experiment.

**Data availability.** Whole-genome sequencing reads for DVT809, DVT1133, DVT1134, and DVT1135 have been deposited in NCBI under BioProject PRJNA876917. Genomes of additional *E. faecalis* strains tested are available at NCBI under BioProject PRJNA649986.

## SUPPLEMENTAL MATERIAL

Supplemental material is available online only.

**SUPPLEMENTAL FILE 1**, PDF file, 0.1 MB.

## ACKNOWLEDGMENTS

This study was funded by grant R03AI168491 from the National Institute of Allergy and Infectious Diseases (D.V.T.) and by the Department of Medicine at the University of Pittsburgh School of Medicine (D.V.T.). Y.L. was supported by a collaborative education and research agreement between the Tsinghua University School of Medicine and the University of Pittsburgh School of Medicine. The funders had no role in study design, data collection and analysis, decision to publish, or preparation of the manuscript.

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
