## [Reviewer comments · Microbiology Spectrum]

Microbiology Spectrum

Carbonic anhydrase inhibition as a target for antibiotic synergy in enterococci

Gayatri Shankar Chilambi, Yu-Hao Wang, Nathan Wallace, Chetachukwu Obiwuma, Kirsten Evans, Yanhong Li, Menna-Allah Shalaby, Daniel Flaherty, Ryan Shields, Yohei Doi, and Daria Van Tyne

Corresponding Author(s): Daria Van Tyne, University of Pittsburgh School of Medicine

Review Timeline:

Submission Date:	September 28, 2022
Editorial Decision:	January 25, 2023
Revision Received:	May 8, 2023
Accepted:	May 19, 2023

Editor: Jose Lemos

Reviewer(s): The reviewers have opted to remain anonymous.

Transaction Report:

DOI: <https://doi.org/10.1128/spectrum.03963-22>

January 25, 2023

Dr. Daria Van Tyne
University of Pittsburgh School of Medicine
3550 Terrace Street
Scaife Hall S853
Pittsburgh, Pennsylvania 15213

Re: Spectrum03963-22 (Carbonic anhydrase inhibition as a target for antibiotic synergy in enterococci)

Dear Dr. Daria Van Tyne:

Thank you for submitting your manuscript to Microbiology Spectrum and my apologies for the long wait. Your manuscript was reviewed by 2 experts in the field. Both reviewers made several suggestions to improve the quality and impact of your study, which will include performing few additional experiments. When submitting the revised version of your paper, please provide (1) point-by-point responses to the issues raised by the reviewers as file type "Response to Reviewers," not in your cover letter, and (2) a PDF file that indicates the changes from the original submission (by highlighting or underlining the changes) as file type "Marked Up Manuscript - For Review Only". Please use this link to submit your revised manuscript - we strongly recommend that you submit your paper within the next 60 days or reach out to me. Detailed instructions on submitting your revised paper are below.

Link Not Available

Sincerely,

Jose Lemos

Journals Department
Reviewer comments:

Reviewer #1 (Comments for the Author):

Reviewer #2 (Comments for the Author):

Enterococcus faecalis-associated endocarditis has a high mortality rate and can be difficult to treat. With the emergence of multidrug-resistant strains, alternative treatment options are much needed. Through in vitro evolution and whole genome sequencing, Chilambi and colleagues identified an alpha-carbonic anhydrase-truncated E. faecalis mutant in a gentamicin hypersusceptible isolate from an endocarditis patient. The authors observed an increased ATP level and membrane permeability in the mutant, which could lead to increased gentamicin uptake and susceptibility. Wild-type strains treated with a carbonic anhydrase inhibitor acetazolamide showed enhanced susceptibility to gentamicin, supporting the finding that alpha-carbonic anhydrase inactivation contributes to enhanced gentamicin killing.

Major comments

1. The authors used a plate assay and Texas Red-labeled gentamicin to measure the uptake of gentamicin and found an increased fluorescent signal in the hypersusceptible strain DVT809. However, the assay condition could be suboptimal: both wild-type (DVT1134) and the susceptible (DVT809) strains were treated with 2.5 ug/mL labeled gentamicin. Given that DVT1134 had a gentamicin MIC of 16 ug/mL and DVT809 had a MIC of 1 ug/mL, the treatment could selectively kill DVT809 and generated more dead/dying cells. Additional data is needed to confirm that the increased signal did not come from dead or dying cells. On top of that, the plate assay could only tell if there are more cells associated with the labeled gentamicin. To examine gentamicin uptake (into its site of action), the authors may need to use microscopy or other tools.
2. Treatment with 2 ug/mL alpha-carbonic anhydrase inhibitor acetazolamide enhanced gentamicin killing by 8-64 fold across different wild-type isolates. It is unclear whether the synergism was carbonic anhydrase-dependent or simply due to an off-target effect. The mutant strain DVT809 needs to be tested. The MICs of acetazolamide alone may also need to be included. Previous reports indicate a MIC of 2 ug/mL for acetazolamide. It is possible that the different wild-type strains were killed by 2 ug/mL acetazolamide to a varying degree and the decreased gentamicin MIC was an additive effect. It can also be useful to test another carbonic anhydrase inhibitor.
3. The authors showed an enhanced gentamicin killing in the presence of heme with the wild-type strain DVT1134, suggesting the link between PMF and gentamicin killing. However, the important connection between alpha-carbonic anhydrase, PMF and gentamicin killing was unanswered. To assess if the increased gentamicin susceptibility in the mutant strain is PMF-dependent, DVT809 should be tested for comparison (Fig 2B).
4. The authors proposed that targeting carbonic anhydrase synergizes with antibiotics. However, only gentamicin was tested across different experiments, and the MICs of ampicillin, tetracycline and ciprofloxacin were unaffected. To determine if it's a gentamicin-or aminoglycoside-specific effect, tobramycin should be examined. It could be helpful to include ampicillin or tetracycline as a control across the assays as well.

Minor comments

1. Is there any growth defect of DVT809? It would be helpful to provide the growth curves for comparison.
2. Line 116: incubated with GTTR in what medium?
3. In Fig 2A, it would be helpful to determine the nM ATP/CFU based on a standard curve.
4. For the PI-based membrane permeability assay, the authors may want to discuss if there are more dead cells in DVT809 cultures.
5. The gentamicin-specific MIC change seems to be contradictory to the hypothesis of increased membrane permeability. This might be worth discussing.

Staff Comments:

Preparing Revision Guidelines

Please return the manuscript within 60 days; if you cannot complete the modification within this time period, please contact me. If

you do not wish to modify the manuscript and prefer to submit it to another journal, please notify me of your decision immediately so that the manuscript may be formally withdrawn from consideration by Microbiology Spectrum.

In the manuscript “Carbonic anhydrase inhibition as a target for antibiotic synergy in enterococci,” the authors report a clinical *Enterococcus faecalis* isolate that has high sensitivity to gentamycin. The authors perform an in vitro evolution experiment to isolate revertants (i.e., increased resistance to) and upon genomic sequencing of three, confirm all possess changes in a carbonic anhydrase gene. The authors use one revertant and the sensitive isolate and perform additional characterization in attempts to conclude the exact mechanism. While interesting, there are some comments regarding the experimental design and conclusions made that would strengthen the findings.

The authors do state their results are with gentamycin and not with additional aminoglycosides? How broad is their findings with other aminoglycosides—even inclusion of MIC data for some other aminoglycosides would broaden and add further significance to their findings.

Additional discussion about carbonic anhydrases—how many are there in most *E. faecalis*? Is this particular one associated with a pathway or is it a prediction based on domain, etc.? This analysis does not need to be overly extensive.

Are there any differences in growth (length of lag, generation time), cfu/OD 600, etc. between the clinical susceptible isolate versus the resistant revertant organisms? This is important control data that can be shown and mentioned within the manuscript.

For gentamycin-uptake, as written, cells used are stationary phase cultures (it appears they resuspended cells in PBS after/for normalization). Stationary-phase *E. faecalis* cells are reduced in ATP levels. However, the ATP level measurements and PI experiments are from exponential phase cells. The authors should present the data from cells that have been grown in the same manner to make accurate comparisons.

For GC-FAME, the cells were grown with shaking, but this appears to be the only experiment with shaking. Are the alterations then in membrane fatty acid content related to differences in oxidative stress levels and not necessarily correlated to gentamycin sensitivity?

Similar to the above, aeration of cultures (so high vs. low oxygen) can impact aminoglycoside uptake/effectivity (and carbonic anhydrases are metalloenzymes). Culture growth should be consistent than for these analyses.

FAME analysis: an increase in unsaturated fatty acids can indicate increase fluidity, however, there is data with *E. faecalis* to suggest that the relationship does not always correlate. Further, the authors also state that this would result in permeability differences. The wording is far too strong to support such a claim: alteration in membrane fatty acid content would impact numerous processes, including membrane protein activity and function.

Have the authors performed a checkerboard assay (and calculated FICI) to demonstrate synergy between gentamycin and acetazolamide?

The authors show in Fig. 2B the addition of heme and its effects on the revertant strain (so aminoglycoside resistant). How did heme impact ATP levels—the authors show data for PMF, but does that correlate? The authors refer to publications from another group, but it is important to include these control experiments for their isolates.

Does heme addition render other *E. faecalis* strains (like OG1RF which is used for some experiments) more sensitive to gentamycin?

Response to Reviews

Spectrum03963-22 Carbonic anhydrase inhibition as a target for antibiotic synergy in enterococci

We thank the editor and reviewers for their comments on our manuscript. We have used the reviewers' comments to revise our study, which we believe is now greatly improved. Changes made to the manuscript are highlighted in yellow in the marked-up copy of the manuscript.

Reviewer #1:

In the manuscript "Carbonic anhydrase inhibition as a target for antibiotic synergy in enterococci," the authors report a clinical *Enterococcus faecalis* isolate that has high sensitivity to gentamycin. The authors perform an in vitro evolution experiment to isolate revertants (i.e., increased resistance to) and upon genomic sequencing of three, confirm all possess changes in a carbonic anhydrase gene. The authors use one revertant and the sensitive isolate and perform additional characterization in attempts to conclude the exact mechanism. While interesting, there are some comments regarding the experimental design and conclusions made that would strengthen the findings.

The authors do state their results are with gentamycin and not with additional aminoglycosides? How broad is their findings with other aminoglycosides—even inclusion of MIC data for some other aminoglycosides would broaden and add further significance to their findings.

- ➔ Thank you for this question. We have expanded our testing to include streptomycin and tobramycin. The trend of hypersusceptibility in the DVT809 strain holds across additional aminoglycosides, and this data is now shown in Table 1 and referenced in Lines 216-219.

Additional discussion about carbonic anhydrases—how many are there in most *E. faecalis*? Is this particular one associated with a pathway or is it a prediction based on domain, etc.? This analysis does not need to be overly extensive.

- ➔ Thank you for this comment. We have included more information about carbonic anhydrase enzymes encoded by *E. faecalis* to the Introduction (Lines 68-70).

Are there any differences in growth (length of lag, generation time), cfu/OD 600, etc. between the clinical susceptible isolate versus the resistant revertant organisms? This is important control data that can be shown and mentioned within the manuscript.

- ➔ We agree with the reviewer that differences in *in vitro* growth between isolates should be documented. Indeed, we observed that the DVT809 isolate grew more slowly than the revertant wild type isolate. This data is now shown in Figure S1 and is referenced in Lines 230-232.

For gentamycin-uptake, as written, cells used are stationary phase cultures (it appears they resuspended cells in PBS after/for normalization). Stationary-phase *E. faecalis* cells are reduced in ATP levels. However, the ATP level measurements and PI experiments are from exponential phase cells. The authors should present the data from cells that have been grown in the same manner to make accurate comparisons.

- Thank you for this comment. Based on our review of the literature it seems that use of stationary-phase cultures is the standard method for measuring gentamicin uptake in bacteria. We acknowledge that in other experiments presented we used exponential phase cells, and we have added this to the Discussion as a limitation of the study (Lines 339-341).

For GC-FAME, the cells were grown with shaking, but this appears to be the only experiment with shaking. Are the alterations then in membrane fatty acid content related to differences in oxidative stress levels and not necessarily correlated to gentamycin sensitivity?

Similar to the above, aeration of cultures (so high vs. low oxygen) can impact aminoglycoside uptake/effectivity (and carbonic anhydrases are metalloenzymes). Culture growth should be consistent than for these analyses.

- We thank the reviewer for bringing this discrepancy to our attention. Cultures for all experiments presented in this manuscript (except MIC testing) were grown at 37°C with shaking. This has been added to the Methods section (Lines 84-86).

FAME analysis: an increase in unsaturated fatty acids can indicate increase fluidity, however, there is data with *E. faecalis* to suggest that the relationship does not always correlate. Further, the authors also state that this would result in permeability differences. The wording is far too strong to support such a claim: alteration in membrane fatty acid content would impact numerous processes, including membrane protein activity and function.

- Thank you for this comment. We have softened the wording in presenting and discussing the FAME analysis results accordingly (Lines 271-275 and 318-320).

Have the authors performed a checkerboard assay (and calculated FICI) to demonstrate synergy between gentamycin and acetazolamide?

- The reviewer poses a valid question. We attempted to perform checkerboard assays to calculate FICI values, however in our hands inhibition with acetazolamide alone was highly variable among wild type *E. faecalis* isolates with intact α -CA genotypes. We did, however, reproducibly observe aminoglycoside sensitization in the presence of a fixed concentration of acetazolamide, which is why we elected to only show that data.

The authors show in Fig. 2B the addition of heme and its effects on the revertant strain (so aminoglycoside resistant). How did heme impact ATP levels—the authors show data for PMF, but does that correlate? The authors refer to publications from another group, but it is important to include these control experiments for their isolates.

- Thank you for this comment. We have modified Fig. 2 to include the same three strains and conditions (DVT1134, DVT1134+heme, and DVT809) on both figure panels, and have updated the Results section accordingly (Lines 249-252).

Does heme addition render other *E. faecalis* strains (like OG1RF which is used for some experiments) more sensitive to gentamycin?

- We tested additional isolates, including OG1RF, and observed that heme exposure did cause increased gentamicin sensitivity (2-8-fold shift in MIC). Because this phenomenon is already documented in the literature (<https://pubmed.ncbi.nlm.nih.gov/4044795/>), we have elected not to include these results to maintain the focus of our study on the role of carbonic anhydrase inhibition in modulating gentamicin susceptibility.

Reviewer #2:

Enterococcus faecalis-associated endocarditis has a high mortality rate and can be difficult to treat. With the emergence of multidrug-resistant strains, alternative treatment options are much needed. Through in vitro evolution and whole genome sequencing, Chilambi and colleagues identified an alpha-carbonic anhydrase-truncated *E. faecalis* mutant in a gentamicin hypersusceptible isolate from an endocarditis patient. The authors observed an increased ATP level and membrane permeability in the mutant, which could lead to increased gentamicin uptake and susceptibility. Wild-type strains treated with a carbonic anhydrase inhibitor acetazolamide showed enhanced susceptibility to gentamicin, supporting the finding that alpha-carbonic anhydrase inactivation contributes to enhanced gentamicin killing.

Major comments

1. The authors used a plate assay and Texas Red-labeled gentamicin to measure the uptake of gentamicin and found an increased fluorescent signal in the hypersusceptible strain DVT809. However, the assay condition could be suboptimal: both wild-type (DVT1134) and the susceptible (DVT809) strains were treated with 2.5 ug/mL labeled gentamicin. Given that DVT1134 had a gentamicin MIC of 16 ug/mL and DVT809 had a MIC of 1 ug/mL, the treatment could selectively kill DVT809 and generated more dead/dying cells. Additional data is needed to confirm that the increased signal did not come from dead or dying cells. On top of that, the plate assay could only tell if there are more cells associated with the labeled gentamicin. To examine gentamicin uptake (into its site of action), the authors may need to use microscopy or other tools.

- We thank the reviewer for raising this important point. It is well known that stationary-phase cultures are often more resistant to aminoglycoside-mediated killing. To show this directly with our strains, we tested whether treatment of stationary-phase DVT809 with 2.5ug/mL gentamicin for 30 minutes (mimicking the conditions of the uptake experiment) caused a dramatic reduction in viable bacteria. We observed a modest reduction in CFU/mL in the culture treated with gentamicin compared with the untreated culture (~25% reduction in CFU/mL). Furthermore, cultures were spun down after gentamicin exposure and fluorescence was measured in only the cell pellet, so lysed bacteria would have been removed before measuring. Finally, because uptake data were normalized to CFU at the beginning of the assay, increased bacterial lysis in the DVT809 culture during the experiment would result in an under-estimation of gentamicin uptake in this strain. For these reasons we do not believe that the increased gentamicin uptake observed in DVT809 is due to more dead/dying cells. We have nonetheless added the possibility of using different approaches to study antibiotic uptake to the Discussion section of the revised manuscript (Lines 338-339).

2. Treatment with 2 ug/mL alpha-carbonic anhydrase inhibitor acetazolamide enhanced gentamicin killing by 8-64 fold across different wild-type isolates. It is unclear whether the synergism was carbonic anhydrase-dependent or simply due to an off-target effect. The mutant strain DVT809 needs to be tested. The MICs of acetazolamide alone may also need to be

included. Previous reports indicate a MIC of 2 ug/mL for acetazolamide. It is possible that the different wild-type strains were killed by 2 ug/mL acetazolamide to a varying degree and the decreased gentamicin MIC was an additive effect. It can also be useful to test another carbonic anhydrase inhibitor.

→ Thank you for this comment. As we stated in the response to Reviewer #1's comment about acetazolamide, in our hands inhibition with acetazolamide alone was highly variable among wild type *E. faecalis* isolates with intact CA genotypes. When we did record MICs, they were in the range of 16-256ug/mL. We have nonetheless added DVT809 to Table 2, which shows that acetazolamide treatment did not affect the gentamicin hypersusceptibility phenotype of that strain.

3. The authors showed an enhanced gentamicin killing in the presence of heme with the wild-type strain DVT1134, suggesting the link between PMF and gentamicin killing. However, the important connection between alpha-carbonic anhydrase, PMF and gentamicin killing was unanswered. To assess if the increased gentamicin susceptibility in the mutant strain is PMF-dependent, DVT809 should be tested for comparison (Fig 2B).

→ Thank you for this comment. We have added DVT809 to Figure 2B.

4. The authors proposed that targeting carbonic anhydrase synergizes with antibiotics. However, only gentamicin was tested across different experiments, and the MICs of ampicillin, tetracycline and ciprofloxacin were unaffected. To determine if it's a gentamicin-or aminoglycoside-specific effect, tobramycin should be examined. It could be helpful to include ampicillin or tetracycline as a control across the assays as well.

→ We agree, and have now included streptomycin and tobramycin testing in Table 1. We also tested whether acetazolamide treatment increased bacterial susceptibility to tetracycline for the isolates shown in Table 1, and did not observe a change in MIC. We have elected not to include this additional data in the manuscript to maintain the focus of the study on how carbonic anhydrase inhibition impacts aminoglycoside susceptibility in *E. faecalis*.

Minor comments

1. Is there any growth defect of DVT809? It would be helpful to provide the growth curves for comparison.

→ Thank you for this comment, which was also brought up by Reviewer #1. We have collected *in vitro* growth curves for DVT809 and DVT1134 in both medias used in the manuscript, and this data is now shown in Figure S1 and referenced in Lines 230-232.

2. Line 116: incubated with GTTR in what medium?

→ Bacteria were grown in MHB and were then incubated in GTTR in PBS for 30 minutes. This is now noted in Lines 135-138.

3. In Fig 2A, it would be helpful to determine the nM ATP/CFU based on a standard curve.

→ Thank you for this comment. We now report the data shown in Figure 2A as pmol ATP per 10⁶ CFU based on a standard curve.

4. For the PI-based membrane permeability assay, the authors may want to discuss if there are more dead cells in DVT809 cultures.

→ Based on our work with these isolates, the PI data and the growth curves shown in Figure S1, we do indeed believe that DVT809 bacteria grown *in vitro* are “sicker” than DVT1134. This is now mentioned in Lines 230-232.

5. The gentamicin-specific MIC change seems to be contradictory to the hypothesis of increased membrane permeability. This might be worth discussing.

→ We now mention the aminoglycoside-specific effects we see in Lines 218-219.

May 19, 2023

Dr. Daria Van Tyne
University of Pittsburgh School of Medicine
3550 Terrace Street
Scaife Hall S853
Pittsburgh, Pennsylvania 15213

Re: Spectrum03963-22R1 (Carbonic anhydrase inhibition as a target for antibiotic synergy in enterococci)

Dear Daria Van Tyne:

Your manuscript has been accepted, and I am forwarding it to the ASM Journals Department for publication. You will be notified when your proofs are ready to be viewed.

Sincerely,

Jose Lemos
Editor, Microbiology Spectrum
